# Critical Damage Values of R200 and 100Cr6 Steels Obtained by Hot Tensile Testing

**DOI:** 10.3390/ma12071011

**Published:** 2019-03-27

**Authors:** Zbigniew Pater, Andrzej Gontarz

**Affiliations:** Faculty of Mechanical Engineering, Lublin University of Technology, Nadbystrzycka 36, 20-618 Lublin, Poland; a.gontarz@pollub.pl

**Keywords:** damage criterion, tensile test, FEM, hot forming of steel

## Abstract

The paper proposes a new method for determining critical damage values in hot forming processes. The method first involves performing tensile tests of axisymmetric samples and then simulating these tests numerically. Simulations are performed by the finite element method in a three-dimensional state of strain, including thermal phenomena occurring in the forming zone. The elaborated method is universal and can be used for different materials. The study is performed for two steel grades, i.e., R200 railway steel and 100Cr6 bearing steel. The results demonstrate that critical damage values strongly depend on the forming temperature.

## 1. Introduction

The plastic deformation of a material may lead to the separation of this material into pieces (fracturing). One can distinguish two different types of fractures:Brittle fracture, where material cohesion is disrupted due to atomic bond breaking;Ductile fracture, where extensive plastic deformation takes place without disrupting material cohesion.

It is worth stressing that the metal fracture is always preceded by plastic deformation. Even when the material undergoes elastic deformation, plastic strains occur near the separation surface.

From the point of view of metal forming, ductile fracture is more significant as it is one of the most common failure patterns in metal forming processes. The occurrence of ductile fracture describes the strain limit of a material in a given metal forming process. It is associated with a change in the material’s energy due to the accumulation of plastic strains leading to fracture, and can be described with the following equation (known as the damage criterion):(1)$$\int_{0}^{\varepsilon_{f}}\Phi \left(\sigma \right)d\varepsilon =C,$$
where Φ(σ) is a function describing the effect of stress on the rate of void formation and coalescence, *ε_f_* is the strain limit, and *C* is the critical value of the damage function.

An overview of the literature [1,2,3,4,5,6,7,8] reveals that there are a number of ductile damage criteria that are based on the function Φ(σ). A selection of these criteria used in this study is listed in Table 1. 

Although the ductile damage criteria given in Table 1 are provided in many commercial computer simulation programs, their practical application requires the knowledge of the critical damage values, *C*. These values are determined by tensile, compression, or torsion testing, using specimens of specially designed shapes (axisymmetric or flat) in order to accelerate material damage.

A survey of the literature on the subject reveals that the damage criterion described by Equation (1) is widely used in cold forming processes for sheet metal products. Given that the damage criterion replicates ductile damage well when the modeled and experimental stresses are similar [3,9,10,11,12], forming limit diagrams are elaborated. This is usually done by the Nakajima test performed in compliance with the relevant EN ISO 12004-2 standard [9,13]. An alternative solution is the use of the Erichsen cupping test [1] or the tensile test for flat-notched specimens of different shapes and sizes [14]. Thereby, obtained data are implemented in specialist simulation software and used for the analysis of forming processes for sheet metals, such as punching, bending, and sheet-metal stamping [2,4,7,9,15,16,17,18,19,20].

The situation is even more complex in the case of metal forming processes, where critical damage values are determined via tensile and compression tests of axisymmetric specimens and are often averaged [10]. It is considerably difficult to determine the moment of fracture during hot compression testing due to the strong thermal radiation that makes it difficult to record the test with a camera [21], leading to lower accuracy of material damage predictions. In light of the above, the number of publications devoted to the problem of fracture in metal forming processes for solids is much lower than those devoted to metal sheet forming. Here, one can distinguish studies devoted to cold forming methods such as forging [22,23,24], extrusion [2], and drawing [25]. With regard to the critical damage values for elevated temperatures, the literature reports results obtained for materials such as AZ31 magnesium alloy with a temperature range of 50 °C–350 °C [1,15], AA6082-T6 aluminum alloy with a temperature range of 400 °C–550 °C [26], Ti40 titanium alloy with a temperature range of 850 °C–1100 °C [21], and 22MnB5 steel grade with a temperature range of 650 °C–800 °C [27]. However, there are no solutions for the hot forming of steel at temperatures exceeding 900 °C.

This paper proposes an innovative method for determining the value of *C* in hot forming conditions. The method involves performing tensile tests of axisymmetric specimens at varying temperatures and then modeling them numerically. The numerical simulations provide stress and strain parameter values which are required for the determination of the critical damage values of the tested material.

## 2. Experimental Tests 

Two steel grades were investigated in this study: R200 railway steel and 100Cr6 bearing steel. Their selection was dictated by the research on innovative techniques for producing balls, conducted at the Lublin University of Technology (Lublin, Poland) [28,29,30]. This research rests on the assumption that balls can be produced either from bearing steel or—in the case of scrap rail recycling—from railway steel by cross-wedge and skew-rolling processes. The chemical composition of R200 and 100Cr6 sheets of steels are given in Table 2 and Table 3, respectively.

### 2.1. Compression Test

The determination of critical damage values requires the investigation of the material’s behavior under high plastic strains, which is described by the so-called flow curves. This can be done via plastometric testing based on compression, tensile, or torsion tests. In this study, compression tests were performed on cylindrical specimens with a diameter of 5 mm and a length of 10 mm. The effect of the strain rate and temperature on the flow stress of the analyzed steel grades was determined via compression tests that were run for three strain rates $\dot{\varepsilon }$: 0.1 s^−1^, 1 s^−1^, and 10 s^−1^ (the values were selected based on the capacity of the testing machine). The tests were run at a temperature, *T*, set to 1000 °C, 1100 °C, and 1200 °C.

The experimental tests were conducted with the DIL805A/D quenching dilatometer (TA Instruments, New Castle, DE USA) with the capability to deform the specimen by compression. This instrument allows the development of isothermal time-temperature-transformation (TTT) and isothermal transformation (IT) diagrams, as well as the determination of phase transitions and deformation properties of the tested materials. The DIL805A/D allows for performing compression tests with constant strain rates ranging from 0.01–15 s^−1^ or with controlled deformation rates from 0.01–150 mm/s. It is also employs the WinTa 6.2 software suite (New Castle, DE USA), which records length of sample, temperature, and compressive force, and, based on the obtained results, calculates other parameters such as effective strain and flow stress. In this work, three measurements were made for every applied forming parameter (described by the strain rate and temperature of the material).

The compression tests provided data about the compressive forces and corresponding variations in the specimen height for the tested temperatures and strain rates. The data computation by well-known equations led to the determination of the relationship between effective plastic stresses and effective strains plotted as flow curves, as shown in Figure 1 and Figure 2.

Based on the obtained experimental data, we also selected function parameters describing the relationship between the flow stresses and the strains, for a temperature range of 1000–1200 °C and a strain rate ranging from 0.1–10 s^−1^ of the tested steel grades. It was assumed that the function would be described by the relation: (2)$$\sigma_{p}=A\varepsilon^{\left(B+CT\right)}e^{\left(D\varepsilon \right)}{\dot{\varepsilon }}^{\left(E+FT\right)}e^{\left(GT\right)},$$
where *ε*—effective strain, no unit; $\dot{\varepsilon }$—strain rate, s^−1^; *T*—temperature, °C; and *A*, *B*, *C*, *D*, *E*, *F*, *G*—coefficients. The coefficients *A–G* were determined by the Generalized Reduced Gradient (GRG2) optimization method implemented into Microsoft Excel. For the purpose of optimization, the objective function Φ*_σ_* was defined as follows:(3)$$\Phi_{\sigma }=\frac{1}{k}\overset{k}{\underset{i=1}{\sum }}\frac{{\left(\sigma_{pt}-\sigma_{pex}\right)}^{2}}{\sigma_{pex}^{2}}\;100\%,$$
where $\sigma_{pt}$—flow stress value calculated based on Equation (2), $\sigma_{pex}$—experimental flow stress value, and *k*—the number of measurement points (resulting from the resolution applied relative to ε, i.e., every 0.05). In effect, it was possible to develop a material model of R200 steel defined as:(4)$$\sigma_{p}=38072.5\varepsilon^{\left(0.272-0.00014T\right)}e^{\left(-0.503\varepsilon \right)}{\dot{\varepsilon }}^{\left(-0.11+0.000245T\right)}e^{\left(-0.00435T\right)},$$
while that of 100Cr6 steel was expressed as:(5)$$\sigma_{p}=596272.4\varepsilon^{\left(0.939-0.00061T\right)}e^{\left(-0.738\varepsilon \right)}{\dot{\varepsilon }}^{\left(-0.943+0.000869T\right)}e^{\left(-0.00629T\right)}.$$

### 2.2. Tensile Test

Critical damage values were determined via tensile testing of axisymmetric specimens under hot forming conditions. The tests were conducted in a Gleeble 3800 thermal-mechanical physical simulator (Postenkill, NY USA). The axisymmetric specimens had a diameter of 10 mm and a length of 116.5 mm and were screwed on both sides (Figure 3). Necking was made in the specimen center in order to facilitate the location of the strain. The test specimens were produced by machining processes, assuming that the tensile test would be run at least three times for every set of tested variables.

The tensile tests were performed with the PocketJaw module for uniaxial compression and tension. Prior to placing them in the simulator, the test specimens were mounted in the jaws and connected to a thermocouple for measuring the material temperature during the test (Figure 4). A K-type thermocouple was used with a measuring accuracy ±0.0075 T. The tensile test was run according to the following pattern—the specimens were heated with the speed of 10 °C/s to the temperature of 1000 °C, 1100 °C, and 1200 °C, respectively; they were kept at the target temperature for 5 s, then subjected to tension with a stroke displacement rate of 5 mm/s until failure, and finally removed from the simulator. During the test, force, specimen temperature in the necking area, and displacement were measured.

The fractured specimens were carefully examined (Figure 5) to determine the elongation values at which the fracture occurred. The results from the tensile tests are listed in Table 4. Given the lack of specimen sizes to perform multiple repeats, the data given in the table are the averaged values of a minimum of three measurements of the tested steel grades and temperatures. The data from Table 4 were then used to determine the critical damage values of the tested steel grades.

## 3. Numerical Analysis 

The critical damage values were calculated numerically by the finite element method. Individual tensile tests for the geometric parameters listed in Table 4 were simulated using the commercial simulation software Simufact Forming (version 15, Hamburg, Germany).

One of the generated tensile test models is shown in Figure 6. The tensile specimen was formed from octagonal finite elements refined in the region of the specimen necking (approximately 40,000 elements were used every time) and remeshing was not implemented.

The simulations used the flow curves for the two tested steel grades, expressed in Equations (4) and (5). The numerical tensile tests were run with the same parameters as those applied in the experimental tests conducted with the Gleeble 3800 simulator. Experimental and numerical results of the maximum forming forces showed a very good quantitative agreement. For example, for the R200 steel, the differences between these forces were 3.83%, 8.33%, and 1.98% for the temperatures 1000 °C, 1100 °C, and 1200 °C, respectively. 

The numerical simulations enabled the examination of the specimen subjected to tension and the determination of the forces, effective stresses, effective strains, strain rates, and temperatures. Figure 7 shows the effective strain rate in the final stage of the tensile test of the R200 steel specimen at 1100 °C. The highest strain rate (1.93 s^−1^) was within the range of strain rates used in plastometric tests (0.1 s^−1^, 1 s^−1^, and 10 s^−1^). This means that the stroke displacement rate of the Gleeble 3800 system’s moving jaw (5 mm/s) was selected correctly.

The necking (undercut) made on the specimen proved to be effective as it created a deformation in the target region (Figure 8), which is vital for the purpose of the numerical analysis. Although, due to the assumption of material homogeneity, the necking procedure on uniform cylindrical specimens is associated with numerical problems. The effective stresses were also the highest in the region of the specimen necking (Figure 9).

The observed location of the highest stress and strain values was significant in terms of the determination of critical damage values. This means that the damage function will have the highest values in this location, too. For this reason, three virtual sensors (distanced from the specimen axis by 0, 1.2, and 2.4 mm) were inserted in the specimen necking region to calculate the values of the parameters (listed in Table 1) necessary for determining ductile damage. The sensors recorded variations in the properties such as strain rate, effective stress, mean stress, and maximum principal stress. It was found that the values calculated at individual points were not significantly different (Figure 10 and Figure 11). For this reason, in a subsequent part of the analysis, critical damage values were calculated using averaged values of the above strain and stress parameters. The final values of these parameters (*σ_i_*, *σ_m_*, *σ*_1_, *ε_f_*) determined in the way mentioned above by tensile testing are listed in Table 5.

## 4. Results and Discussions 

The numerical results and experimental findings led to the determination of critical damage values (from *C*_1_ to *C*_7_—calculated in accordance with Table 1), as listed in Table 6. When calculating the critical damage according to Oyane’s criterion, based on [7], the material constant was set to *A* = 0.424.

An analysis of the data given in Table 6 demonstrates that the critical damage values depend on the tested steel grade. In all investigated cases, the critical damage values of R200 railway steel were higher than those obtained under the same conditions for 100Cr6 bearing steel. This undoubtedly resulted from the differences in the strength and the hot working properties of the tested materials, as demonstrated by the plastometric tests.

The results demonstrate that increasing the temperature leads to a decrease in the critical damage value. However, the decrease in the *C* value was not identical in all tested cases. The criteria developed by Frudenthal and Zhan et al. were found to be twice as sensitive to temperature as compared to the other analyzed criteria. Given the heavy dependence of the critical damage value on temperature, material fracture in hot working conditions cannot be accurately predicted based on the averaged value of the critical damage obtained in a wider temperature range. In such a case, it is necessary to employ solutions that allow for linking critical damage function values to temperature. This can be a serious constraint when using commercial computer programs that restrict the input to only one critical value of the damage function, a problem which will require establishing additional computational procedures. 

## 5. Conclusions 

Based on the obtained results, the following conclusions can be formulated:Critical damage values in hot working conditions can be determined by a tensile test of notched round bars;The employed method for determining critical damage values is highly universal, and hence can be employed in hot forming processes for all metals and alloys;Critical damage values depend on the steel grade, particularly its strength properties and hot-workability;Critical damage values depend on the forming temperature; The critical damage values obtained for R200 and 100Cr6 steel grades for the temperature range of 1000–1200 °C are listed in Table 6.

## Figures and Tables

**Figure 1 materials-12-01011-f001:**
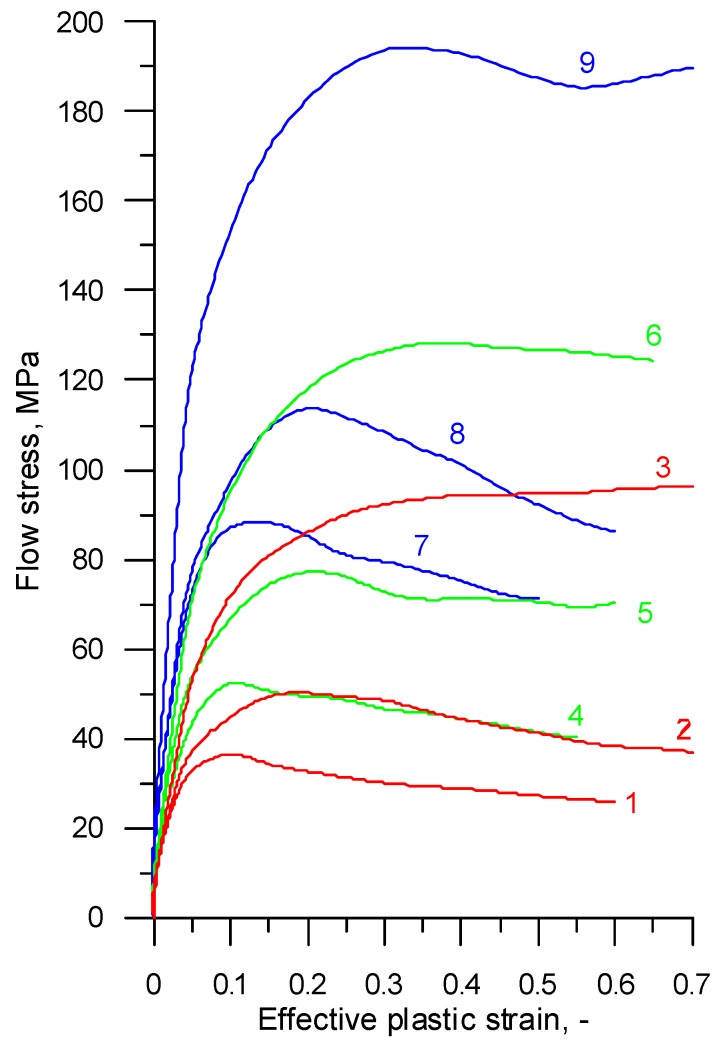
Experimental flow curves for R200 steel, where 1—*T* = 1200 °C, $\dot{\varepsilon }$ = 0.1 s^−1^; 2—*T* = 1200 °C, $\dot{\varepsilon }$ = 1 s^−1^; 3—*T* = 1200 °C, $\dot{\varepsilon }$ = 10 s^−1^; 4—*T* = 1100 °C, $\dot{\varepsilon }$ = 0.1 s^−1^; 5—*T* = 1100 °C, $\dot{\varepsilon }$ = 1 s^−1^; 6—T = 1100 °C, $\dot{\varepsilon }$ = 10 s^−1^; 7—*T* = 1000 °C, $\dot{\varepsilon }$ = 0.1 s^−1^; 8—*T* = 1000 °C, $\dot{\varepsilon }$ = 1 s^−1^; 9—*T* = 1000 °C, $\dot{\varepsilon }$ = 10 s^−1^.

**Figure 2 materials-12-01011-f002:**
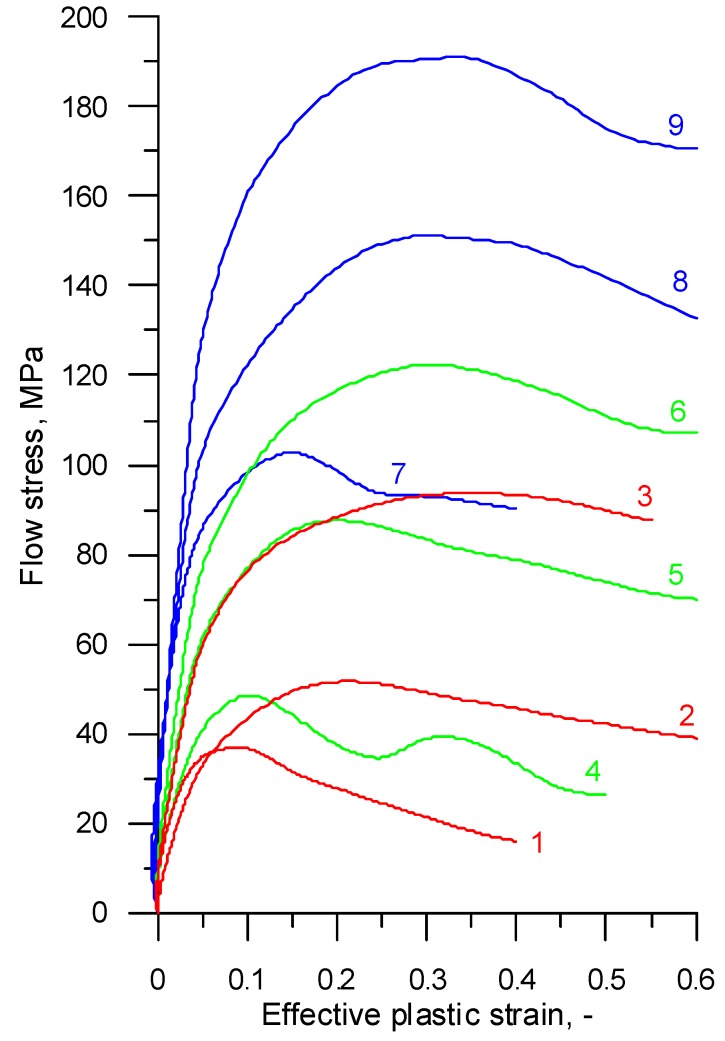
Experimental flow curves for 100Cr6 steel, where 1—*T* = 1200 °C, $\dot{\varepsilon }$ = 0.1 s^−1^; 2—*T* = 1200 °C, $\dot{\varepsilon }$ = 1 s^−1^; 3—*T* = 1200 °C, $\dot{\varepsilon }$ = 10 s^−1^; 4—*T* = 1100 °C, $\dot{\varepsilon }$ = 0.1 s^−1^; 5—*T* = 1100 °C, $\dot{\varepsilon }$ = 1 s^−1^; 6—T = 1100 °C, $\dot{\varepsilon }$ = 10 s^−1^; 7—*T* = 1000 °C, $\dot{\varepsilon }$ = 0.1 s^−1^; 8—*T* = 1000 °C, $\dot{\varepsilon }$ = 1 s^−1^; 9—*T* = 1000 °C, $\dot{\varepsilon }$ = 10 s^−1^.

**Figure 3 materials-12-01011-f003:**
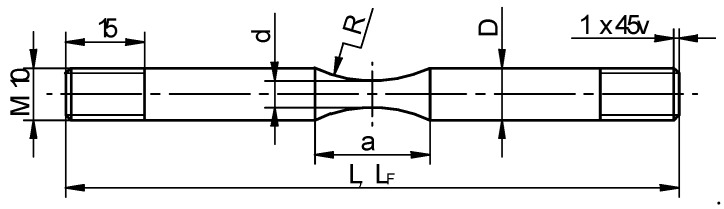
The shape of the tensile specimen and its basic dimensions expressed in mm.

**Figure 4 materials-12-01011-f004:**
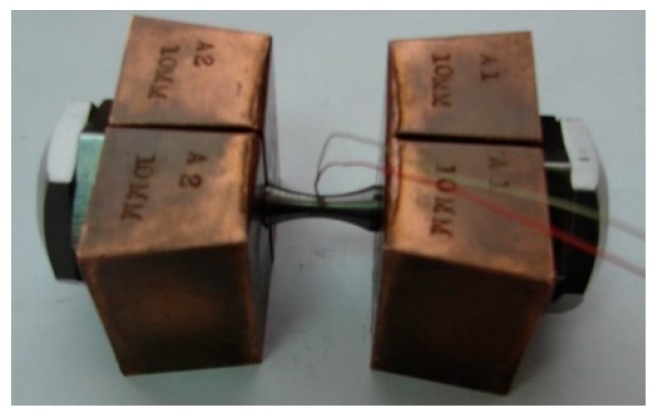
Clamped specimen with a connected thermocouple, prepared for mounting in the Gleeble simulator.

**Figure 5 materials-12-01011-f005:**
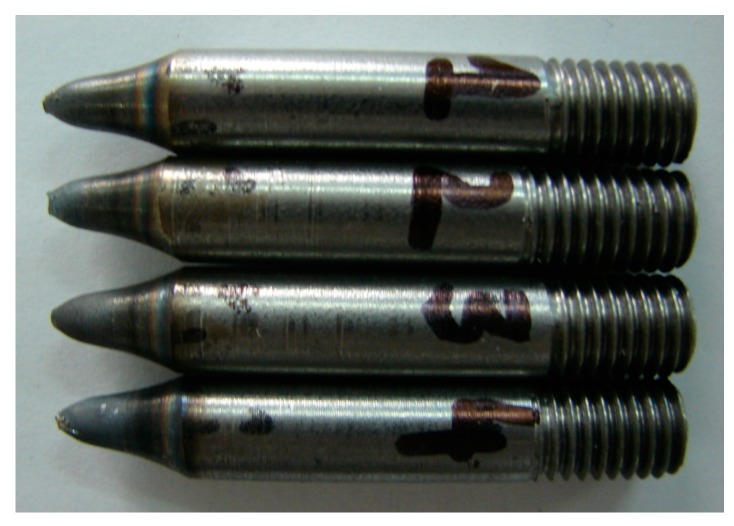
R200 steel specimens fractured in the tensile test performed with the Gleeble 3800 simulator at a temperature range of 1—900 °C (this temperature range was excluded from the fracture analysis since the plastometric test was constrained to the temperature range of 1000 °C–200 °C), 2—1000 °C, 3—1100 °C, and 4—1200 °C.

**Figure 6 materials-12-01011-f006:**
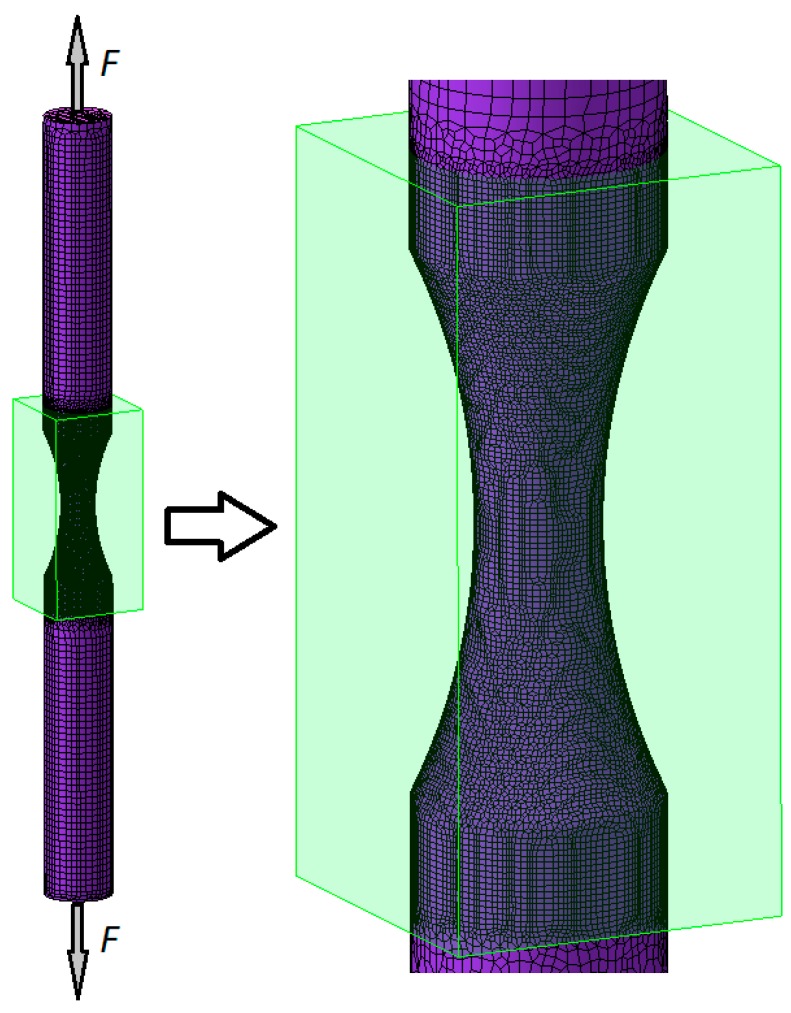
Tensile test model generated in Simufact Forming showing the division of the specimen into finite elements.

**Figure 7 materials-12-01011-f007:**
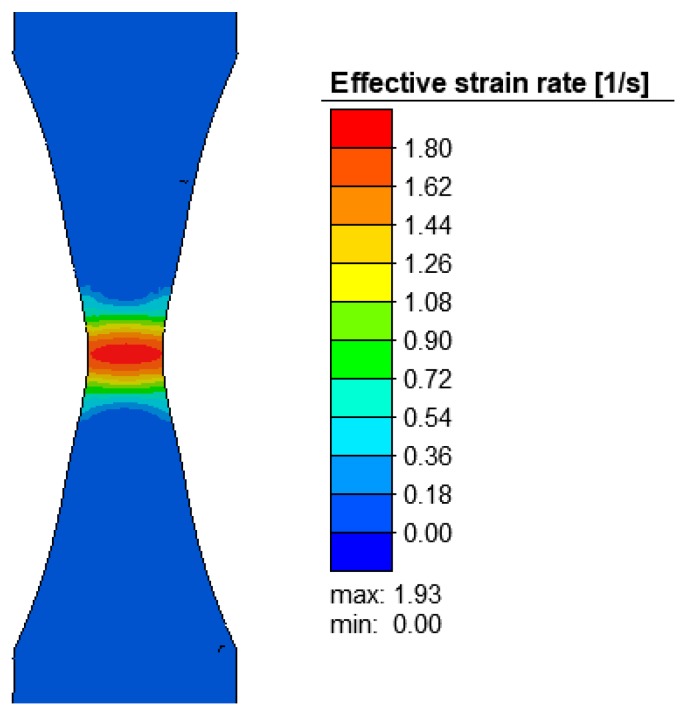
Effective strain rate (in s^−1^) in the axial section of the R200 steel sample under the tensile test at 1100 °C.

**Figure 8 materials-12-01011-f008:**
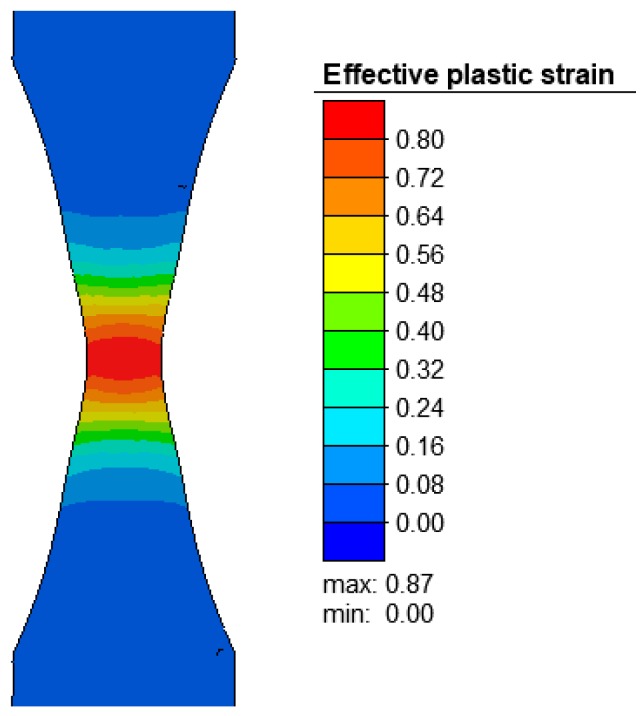
Effective plastic strain in the axial section of the R200 steel sample under the tensile test at 1100 °C.

**Figure 9 materials-12-01011-f009:**
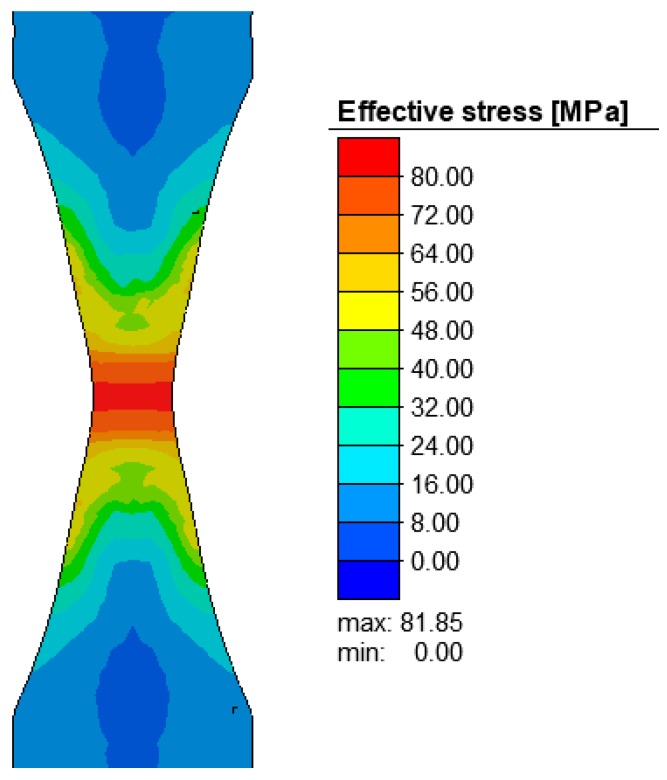
Effective stress (MPa) in the axial section of the R200 steel sample under the tensile test at 1100 °C.

**Figure 10 materials-12-01011-f010:**
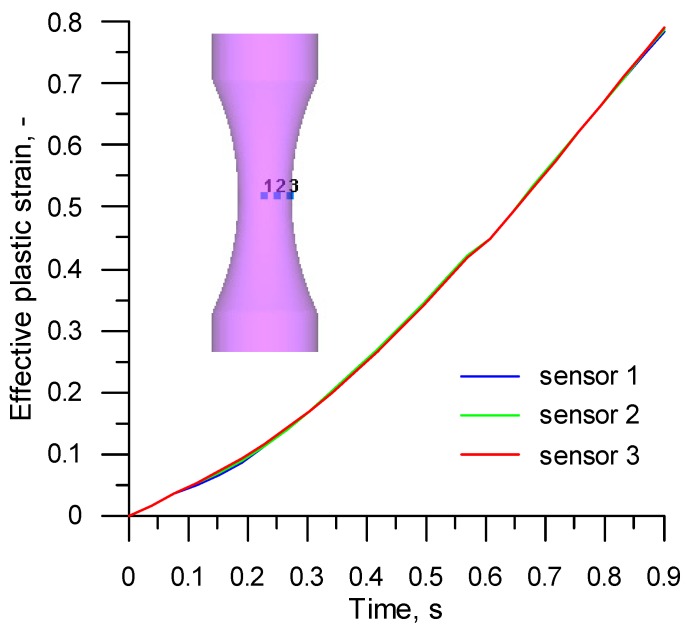
Variations in the effective plastic strain measured by the sensors located in the R200 steel sample under the tensile test at 1100 °C.

**Figure 11 materials-12-01011-f011:**
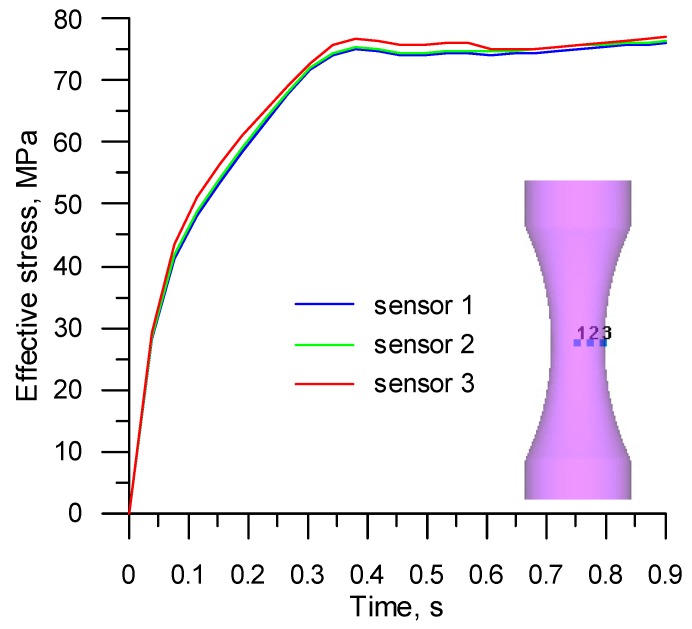
Variations in the effective stress (MPa) measured by the sensors located in the R200 steel sample under the tensile test at 1100 °C.

**Table 1 materials-12-01011-t001:** Selected ductile fracture criteria used in the analysis [1,2,3,4,5,6,7,8].

Criterion (year)	Formula
Freudenthal (1950)	$\int_{0}^{\varepsilon_{f}}\sigma_{i}d\varepsilon =C_{1}$
Cockcroft and Latham (1968)	$\int_{0}^{\varepsilon_{f}}\frac{\sigma_{1}}{\sigma_{i}}d\varepsilon =C_{2}$
Rice and Tracey (1969)	$\int_{0}^{\varepsilon_{f}}exp\left(\frac{3}{2}\frac{\sigma_{m}}{\sigma_{i}}\right)d\varepsilon =C_{3}$
Brozzo et al. (1972)	$\int_{0}^{\varepsilon_{f}}\frac{2\sigma_{1}}{3\left(\sigma_{1}-\sigma_{m}\right)}d\varepsilon =C_{4}$
Oyane (1972)	$\int_{0}^{\varepsilon_{f}}\left(1+A\frac{\sigma_{m}}{\sigma_{i}}\right)d\varepsilon =C_{5}$
Ayada (1984)	$\int_{0}^{\varepsilon_{f}}\frac{\sigma_{m}}{\sigma_{i}}d\varepsilon =C_{6}$
Zhan et al. (2009)	$\int_{0}^{\varepsilon_{f}}\left(\sigma_{1}-\sigma_{m}\right)d\varepsilon =C_{7}$

$\varepsilon_{f}$—critical plastic strain at fracture, $\sigma_{i}$—effective stress, $\sigma_{m}$—mean stress, $\sigma_{1}$—maximal principal stress, *A*—material constant, and *C*—the critical value of the damage function.

**Table 2 materials-12-01011-t002:** The chemical composition of R200 steel (weight percentage, %).

C	Mn	Si	P	Cr	S	Al	N	V	Fe
0.38–0.62	0.65–1.25	0.13–0.60	≤0.04	≤0.15	≤0.035	≤0.004	≤0.01	≤0.03	balance

**Table 3 materials-12-01011-t003:** The chemical composition of 100Cr6 steel (weight percentage, %).

C	Mn	Si	P	S	Cr	Ni	Cu	Ni+Cu	Fe
0.95–1.1	0.25–0.45	0.15–0.35	≤0.027	≤0.020	1.3–1.65	≤0.30	≤0.30	≤0.50	balance

**Table 4 materials-12-01011-t004:** Dimensions of the tensile specimens used for determining damage function critical values (denoted in accordance with Figure 3).

Steel Grade	*T*, °C	*D*, mm	*d*, mm	*a*, mm	*L*, mm	*L_F_*, mm
R200	1000	9.97	5.10	21.93	116.70	120.77
	1100	9.95	5.09	21.85	116.55	121.05
	1200	10.00	4.99	21.88	116.72	119.75
100Cr6	1000	10.00	5.00	21.92	116.40	120.25
	1100	10.04	5.18	21.98	116.31	120.32
	1200	9.90	5.05	22.12	116.58	119.34

**Table 5 materials-12-01011-t005:** Final values of selected parameters used in the tensile testing of R200 and 100Cr6 steels in the temperature range of 1000–1200 °C.

Parameter	R200			100Cr6		
	*T* = 1000 °C	*T* = 1100 °C	*T* = 1200 °C	*T* = 1000 °C	*T* = 1100 °C	*T* = 1200 °C
*ε_f_*, -	0.929	0.854	0.540	0.633	0.518	0.410
*σ*_1_, MPa	118.87	86.70	54.72	149.48	75.81	51.84
*σ_i_*, MPa	106.11	81.30	53.21	143.04	75.70	52.16
*σ_m_*, MPa	48.16	32.50	19.27	54.15	25.35	17.07

**Table 6 materials-12-01011-t006:** Critical values of the damage function for R200 and 100Cr6 steels in the temperature range of 1000–1200 °C, according to the criteria listed in Table 1.

Criterion	R200			100Cr6		
	*T* = 1000 °C	*T* = 1100 °C	*T* = 1200 °C	*T* = 1000 °C	*T* = 1100 °C	*T* = 1200 °C
Freudenthal *C*_1_	91.697	61.100	25.094	82.734	35.797	19.689
Cockcroft and Latham *C*_2_	1.010	0.895	0.563	0.652	0.529	0.418
Rice and Tracey *C*_3_	1.148	1.033	0.652	0.759	0.618	0.487
Brozzo et al. *C*_4_	1.013	0.894	0.564	0.651	0.528	0.414
Oyane *C*_5_	1.095	0.992	0.626	0.730	0.596	0.470
Ayada *C*_6_	0.392	0.325	0.203	0.229	0.183	0.142
Zhan et al. *C*_7_	61.140	40.880	16.793	55.401	23.996	13.263

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
