# Peer review of "Critical Damage Values of R200 and 100Cr6 Steels Obtained by Hot Tensile Testing"

_materials, 2019, doi:10.3390/ma12071011_

Reviewer 1 Report

In this work, a new method for determining the value of C in hot forming conditions has been introduced. For this method, some tensile tests of axisymmetric specimens at different temperatures have been done and then model them numerically. Before further consideration the following issues should be considered and addressed:

1.      In table 1 and 2, the unit of the chemical composition should be given.

2.      Line 78-80 should be moved to the introduction section.

3.      has the friction between the sample surface and anvils during the hot compression test been considered? How?

4.      In Fig.1 and 2, why the effective strains of some conditions are different? i.e. in Fig.1,  condition 1,2,7!

5.      How the Eq.9 and 10 were achieved? Describe more the procedure of calculation.

6.      The heading of ‘Results’ should be ‘Results and Discussions’

7.      Has the fracture surface of samples been analyzed? If yes please add and discuss.

Author Response

Thank you very much for all remarks and suggestions.

In this work, a new method for determining the value of C in hot forming conditions has been introduced. For this method, some tensile tests of axisymmetric specimens at different temperatures have been done and then model them numerically. Before further consideration the following issues should be considered and addressed:

1. In table 1 and 2, the unit of the chemical composition should be given.

The units have been added to the description of the tables.

2. Line 78-80 should be moved to the introduction section.

In our opinion, moving lines 78-80 to the introduction section will have an unfavorable effect. For this reason, we have decided not to follow the Reviewer’s suggestion.

3. has the friction between the sample surface and anvils during the hot compression test been considered? How?

The flow stress was determined automatically by the DIL 805 A/D dilatometer software. According to the machine manual, this software takes account of friction, yet the manual does not specify how and based on what relations it is done.

4. In Fig.1 and 2, why the effective strains of some conditions are different? i.e. in Fig.1,  condition 1,2,7!

In general, the flow curves were determined for the effective strain amounting up to 0.6. If possible, we tried to achieve a higher effective strain. However, in some cases, the computer program would finish the deformation process earlier. The test was repeated three times for each case. Figs. 1 and 2 show the obtained results in a reliable way.

5.  How the Eq.9 and 10 were achieved? Describe more the procedure of calculation.

The description of the procedure has been extended, and Eqs. (9) and (10) have now become Eqs. (11) and (12).

6.  The heading of ‘Results’ should be ‘Results and Discussions’

The heading has been corrected as suggested by the Reviewer.

7.   Has the fracture surface of samples been analyzed? If yes please add and discuss.

The fracture surface of samples has not been analyzed. The objective of the work was to determine the critical value of the damage function which could be implemented to computer simulation programs used for analyzing hot forming processes, in particular cross and skew rolling.  

Reviewer 2 Report

Review of the manuscript "Critical Damage Values of R200 and 100Cr6 Steels Obtained by Hot Tensile Testing" by the authors Zbigniew Pater and Andrzej Gontarz.

This is an interesting work however is require major improvements. 

Some comments to the authors:

Figure 1 &2 presents the Experimental flow curves of studied material...it is noted that the strength increases with strain rate, however in Figure 1 the difference between curve 1 and 9 in terms of effective plastic strain is not obvious, the same in Figure 2 when compare curve 2 with 9. Please very and better explain this results from these two curves, in order to well present the physical meaning.

The samples from Figure 3 is related to any standard ? or just machine capabilities ?

Please give details of thermocouple accuracy and how was calibrated.

Fig. 5 do not bring evident details of better elongation between samples please revise it.

May be the Table 4 should be inserted before Figure 1 is little bit confused in this manner. 

Why was used octagonal finite elements ?

Can you provide details of difference between numerical/experimental results in terms of accuracy.

Figure 7,8, 9 should belongs to results section.

Why the the material constant was set equal to  A=0.424.  ?

The line from 190 to 195 requires better explanation.

This phrase doesn't make sense "The C function values obtained for the temperature of 1200 °C considerably differ from those obtained for the other two temperatures, which can be explained by reduced hot-workability of the material due to excessive grain growth." please revise it. 

The conclusion should be reformulate:

such as "critical damage values for high temperature ranges can be determined by tensile testing and FEM-based numerical analysis" is not very obvious.

this is somehow evident "critical damage values strongly depend on the forming temperature "

Therefore please revise carefully the entire paper and especially conclusion part.

Author Response

Thank you very much for all remarks and suggestions.

Review of the manuscript "Critical Damage Values of R200 and 100Cr6 Steels Obtained by Hot Tensile Testing" by the authors Zbigniew Pater and Andrzej Gontarz.

This is an interesting work however is require major improvements. 

Some comments to the authors:

Figure 1 &2 presents the Experimental flow curves of studied material...it is noted that the strength increases with strain rate, however in Figure 1 the difference between curve 1 and 9 in terms of effective plastic strain is not obvious, the same in Figure 2 when compare curve 2 with 9. Please very and better explain this results from these two curves, in order to well present the physical meaning.

Figs. 1 and 2 show the reliably-determined experimental curves of the flow stress-effective strain and strain rate-temperature relations. Generally, the curves were determined based on the assumption that the maximum strain value in the test is 0.6. However, given the future application of thereby developed material model, the strain value – if possible – was  increased. In some cases, the dilatometer-managing program would finish the test earlier.  Hence, the differences between the strain values obtained in individual tests. This fact, however, posed no obstacle to the objective of our study, i.e. the development of material models of the analyzed steel grades. A description of the development of these models has been added to the manuscript.

The samples from Figure 3 is related to any standard ? or just machine capabilities ?

The shape, dimensions and mounting of the samples were imposed by the test stand. The shape of sample necking could have, naturally, been altered. The employed sample shape was supposed to facilitate the detection of strain, which was of vital importance particularly in numerical calculations. In ideal condition of theoretical calculations, a constant diameter sample would uniformly elongate over the entire measuring length.

Please give details of thermocouple accuracy and how was calibrated.

Due to the short duration of the test, a K-type thermocouple was used, its measurement accuracy according to DIN EN 60584-2 being ±0.0075T. Therefore, for the highest temperature, the thermocouple accuracy was  ±9°C. The thermocouple was calibrated by means of an electronic standard. This information has been added in the manuscript.

Fig. 5 do not bring evident details of better elongation between samples please revise it.

Fig. 5 is only an explanatory figure. As regards the strain in sample necking, it is best illustrated by the parameter εf­ describing the strain at failure. The values of this parameter have been reported in new Tab. 5 added in the manuscript.

May be the Table 4 should be inserted before Figure 1 is little bit confused in this manner. 

Table 4 has been inserted after Fig.1.

Why was used octagonal finite elements ?

The selection of finite elements was dictated by the employed computer simulation software. The simulation program Simufact.Forming performs much better when using octagonal rather than tetragonal finite elements. This observation is confirmed by the authors’ experience, as they have been using this computer program (and its predecessors) for over 20 years.

Can you provide details of difference between numerical/experimental results in terms of accuracy.

Given the cost of the tests, only 3 tests could be performed for the tested parameters. Due to the small number of tests, a statistical analysis of the results was omitted. In terms of comparison between numerical and experimental results, we can only compare the tensile forces. For the steel grade R200 this comparison is as follows. For the temperature of 1000 °C the maximum tensile force obtained in the experiments was 2.09 kN, while the numerical force was 2.01 kN. For the temperature of 1100 °C, the experimental and numerical forces were 1.32 kN and 1.43 kN, respectively, while for T=1200 °C – 1.01 kN and 1.03 kN. This means that the average accuracy between the numerical and experimental results of the forces for this steel grade is 4.7%. This information has been added in the manuscript.

Figure 7,8, 9 should belongs to results section.

In principle, the Reviewer is right. Nevertheless, given the objective of the study, i.e. determination of the critical values of the damage functions, we decided to leave the figures in the numerical analysis section. In other words, moving the figures in question to the results section would make the numerical analysis section look marginal.

Why the the material constant was set equal to  A=0.424.  ?

The A parameter value was taken after the study by Hambli and Reszka – Ref. [7] in the References section.

The line from 190 to 195 requires better explanation. This phrase doesn't make sense "The C function values obtained for the temperature of 1200 °C considerably differ from those obtained for the other two temperatures, which can be explained by reduced hot-workability of the material due to excessive grain growth." please revise it. 

The manuscript has been revised.

The conclusion should be reformulate:

such as "critical damage values for high temperature ranges can be determined by tensile testing and FEM-based numerical analysis" is not very obvious.

this is somehow evident "critical damage values strongly depend on the forming temperature "

Therefore please revise carefully the entire paper and especially conclusion part.

The conclusion section has been revised.

Reviewer 3 Report

I hardly agree with the term "plastic stress" by the physical sense of the words used. I would prefer "flow stess" or simply "stress".

The main question: why flow curves were obtaind in compression, but tests and calculations for damage evaluations were conducted in tension?

Is it really important by some reasons, or it was just a case?

The authors do not provide the value of εf needed to calculate Ci values in Tab. 5 according to Eqs. (2)-(8). 

They must indicate that value and give some explanation on its choice.

And finally, it is interesting how the authors succeeded to perform simulations with descending flow curves like in Figs (1) and (2)?

It would be useful to read some explanations of the problem in the paper.

Page 3, line 75 In Table 2 the content of Si is twice indicated.

Page 3, line 92 "software suite that measures" I'd say that software acquires, obtaines or gains something, but it does not measure.

Page 3, line 93 How can a software calculate strain without gaining changes in dimensions (length)?

Pages 3-4, Eqs. (9) and (10): The authors have to indicate the dimensions of the stress, strain, strain rate and temperature in those equations, because they have numbers which significantly depend on the dimensions used.

Page 4, line 113 Dimensions in mm?  - should be clarified.

Page 5, line 119  Displacement rate should have dimension of m/s or mm/s but not mm.

Page 5, line 134 Definitly, it must be reference to Fig.3 but not to Fig.1

Page 6, lines 142-143 There is some disagreement. The authors write about mesh refinement at line 142 and then that "Mesh refining was not implemented." They should explain that.

Page 6, line 147: Please clarify, the flow curves from figures 1,2 or from equations (9), (10) were used.

Page 6, line 151: My opinion is that simulations enable not the measurement but the calculation of something.

Page 6, line 153 Strain rate and displacement rate have different dimensions, so they can not be compared.

Page 6, line 154 The dimensions for the displacement rates are indicated incorrectly.

Page 6, lines 163-165 The authors should clarify whether the sensors were virtual (used in simulations) or real (in experiments). Though this is the section devoted to simulations, but the words "sensors", "mounted" and "measured" sound like a real experiment.

Page 8, line 193 "that that" - one "that" should be removed

Author Response

Thank you very much for all remarks and suggestions.

I hardly agree with the term "plastic stress" by the physical sense of the words used. I would prefer "flow stress" or simply "stress".

The term has been changed to “flow stress”.

The main question: why flow curves were obtained in compression, but tests and calculations for damage evaluations were conducted in tension? Is it really important by some reasons, or it was just a case?

The methods were selected on purpose. The models of materials were supposed to be used for modeling rolling processes in which the strains are high. For this reason, we decided to use a compression test to determine flow curves, as the strains obtained in this test are several times higher than those obtained by tensile testing.  On the other hand, we decided to select a tensile test for determining critical damage functions due to the fact that it is difficult to detect circumferential cracks in samples deformed under hot forming conditions because thermal radiation makes it difficult to record such test with a camera.

The authors do not provide the value of εf needed to calculate Ci values in Tab. 5 according to Eqs. (2)-(8). They must indicate that value and give some explanation on its choice.

The value has been added in Tab. 5 which lists the final values of the parameters used to calculate C­i values.

And finally, it is interesting how the authors succeeded to perform simulations with descending flow curves like in Figs (1) and (2)? It would be useful to read some explanations of the problem in the paper.

The obtained flow curves were used to develop Eqs. (11) and (12) describing the flow stress-temperature and effective strain-strain rate relations. These equations were used in the numerical calculations. The manuscript has been extended to include a description of developing the above equations.

Page 3, line 75 In Table 2 the content of Si is twice indicated.

The table has been corrected.

Page 3, line 92 "software suite that measures" I'd say that software acquires, obtaines or gains something, but it does not measure.

The word has been replaced with “records”.

Page 3, line 93 How can a software calculate strain without gaining changes in dimensions (length)?

The word “changes” was replaced with “length changes”.

Pages 3-4, Eqs. (9) and (10): The authors have to indicate the dimensions of the stress, strain, strain rate and temperature in those equations, because they have numbers which significantly depend on the dimensions used.

The manuscript has been extended to include a description of studies involving the development of equations describing the flow stresses of investigated materials. The Reviewer’s suggestion has been included in this description.

Page 4, line 113 Dimensions in mm?  - should be clarified.

The dimension has been corrected.

Page 5, line 119  Displacement rate should have dimension of m/s or mm/s but not mm.

The dimension has been changed to mm/s.

Page 5, line 134 Definitly, it must be reference to Fig.3 but not to Fig.1

The error has been corrected.

Page 6, lines 142-143 There is some disagreement. The authors write about mesh refinement at line 142 and then that "Mesh refining was not implemented." They should explain that.

It has been changed to “Remeshing was not implemented”.

Page 6, line 147: Please clarify, the flow curves from figures 1,2 or from equations (9), (10) were used.

The simulation was performed using the material models described with Eqs. (9) and (10), after the changes – they have become Eqs. (11) and (12).

Page 6, line 151: My opinion is that simulations enable not the measurement but the calculation of something.

The word in question has been changed to “determination”.

Page 6, line 153 Strain rate and displacement rate have different dimensions, so they cannot be compared.

The Reviewer is right, in both cases it should be “strain rate”.

Page 6, line 154 The dimensions for the displacement rates are indicated incorrectly.

Following the Reviewer’s suggestion, the above has been corrected.

Page 6, lines 163-165 The authors should clarify whether the sensors were virtual (used in simulations) or real (in experiments). Though this is the section devoted to simulations, but the words "sensors", "mounted" and "measured" sound like a real experiment.

The sensors were virtual. The words in question have been replaced with more adequate ones.

Page 8, line 193 "that that" - one "that" should be removed

The redundant word has been removed.

Round  2

Reviewer 1 Report

The revision is satisfactory and so this paper can be accepted in this form.

Reviewer 2 Report

Dear Authors, 

Thank you for revised paper that responded to my questions.

Therefore, I advice acceptance in the present form.